# Do We Need New Crops for Arid Regions? A Review of Fruit Species Domestication in Israel

**Yosef Mizrahi**

Department of Life Sciences, Ben-Gurion University of the Negev, Beer Sheva 844190, Israel; mizrahi@bgu.ac.il

**Abstract:** Climatic changes have created the imminent need for the development of new crops for arid regions. We started to domesticate and introduce wild and exotic fruit trees to our deserts in 1984. We tested different species in five eco zones in Israel, differing from each other in terms of maximum and minimum temperatures, type and degrees of salinities, water evaporation rates, rainfall, etc. We succeeded in cultivating pitaya species using hybrids from the *Hylocereus* and *Selenicereus* genera, and with a different species from the Cactaeae *Cereus peruvianus*, which we named Koubo. These two species are from the Cactaceae family, known for high water use efficiencies (WUE). We already have investors who started the semi-commercial production of Marula, *Sclerocarya birrea* sbsp. *Caffra*, and Argan—*Argania spinosa*. In spite of the fact that we developed good clones and knowledge of how to grow and ship White Sapote, *Casimiroa edulis*, and Indian jujube, *Ziziphus mauritiana*, we failed due to a lack of marketing research and development, which is essential for such a project. We have gene banks of some other new fruit crops waiting for investors to grow and bring them into the domestic and world markets.

**Keywords:** domestication; genetics; physiology; drought; heat; salinity

## 1. Introduction

Climatic changes have created an imminent need for the development of new crops for arid regions, in which common crops will likely fail in the future due to lack of water and very high temperatures. Israel is a semi-arid and arid country and hence an ideal place to develop this kind of new future crop. Israel is also a very small country with a small domestic market, so our research and development (R&D) efforts need to be channeled into the global market. We cannot compete in the world market with large production volumes; hence, the exotic niche is most attractive for us. This means not channeling our R&D efforts towards conventional crops but towards crops of the future.

We aim at developing proper knowledge of how to grow and market these new crops, which do not exist in the world market. Several publications have shown that there are many wild plants which were/are used by humans that have been neglected by the scientific community. These plants deserve R&D as crops for the future [1–3].

With the help of the ethnobotanist Dr James Alon Aronson, we prepared a list of fruit tree species either wild or grown only by indigenous people (of dry regions from Australia, Africa, America, and Asia) from the following families: Apocynaceae, Anacardiaceae, Bombacaeae, Cactaceae, Caesalpiniaceae, Ebenaceae, Euphorbiaceae, Flacourtiaceae, Loganiaceae, Rhamnaceae, Rubiaceae, Santalaceae, and Sapotaceae (see Appendix A) All these taxons selected for our R&D program from regions of warm and/or dry climates. Seeds were collected in their country of origin from various locations to enable maximum genetic diversity to increase the chance that some of them will excel under our conditions.

The Negev and Judean deserts of Israel are close to our institute, in which different locations with different climatic and water salinity characteristics exist (Table 1).

**Table 1.** Environmental and other characteristics (average values) of the introduction sites in the Israeli deserts.

| Locations | Besor | Ramat Negev | Neot Hakikar | Qetura | Gilgal |
|---|---|---|---|---|---|
| Maximum temperature °C | | | | | |
| January (coldest month) | 17/8 | 16/3 | 20/10 | 19/5 | 19/5 |
| August (hottest month) | 32/20 | 35/18 | 40/25 | 39/23 | 39/23 |
| Pan evaporation (mm/year) | 1900 | 2300 | 3900 | 3600 | 3600 |
| Rainfall (mm/year) | 200 | 90 | 50 | 30 | 80 |
| Soil texture | sandy-loam | loam | sandy-loam | sandy-loam | sandy-loam |
| Irrigation water | | | | | |
| EC (dS/m) | 1.0 | 1.0–4.0 | 4.0 | 4.0 | 0.5 |
| pH | 7.3 | 7.3 | 7.4 | 7.6 | 7.6 |

The salinity in Qetura is mainly from Ca, Mg, and $SO_4$, while in Neot-Hakikar and Ramat Negev it is mainly Na and Cl. We also planted some species in the Gilgal (Judean desert), which has similar conditions to those in Qetura but with excellent water properties; EC = 0.5 dS/m.

Seeds of species from the above-mentioned genera were germinated in quarantine and, after 6 months of inspection for pests, cleared and transferred to our nurseries. We established introduction orchards in the locations specified in Table 1. As mentioned, the locations differed in their water, climatic, and soil conditions. We started as early as 1984 [4,5]. We planted 15 seedlings (in three replications, 5 of each) in the five introduction orchards differing from each other in their average maximum and minimum temperatures, annual rainfall, pan evaporation rates, and water salinity. Soil salinity is not important, since it can be leached to the degree of salinity in the irrigation water. All introduction orchards were irrigated with around 500 mm per year, since not enough rain falls in these locations. In the Ramat Negev site, we had both a good water quality of E.C. = 1 dS/m and saline water E.C. = 4 dS/m; hence, we planted 30 seedlings from each species. After orchard establishment, we followed their growth; time to first flower production; reproductive system [6]; and quality of fruits in terms of appearance, size, taste, and shelf life. Many of the species did not survive in all the orchards, and some only in the Besor, which possesses the best horticultural characteristics (Table 1). Some died due to salinity and some others due to extreme temperatures. Once a species was doing well, we started to select the best genotypes from the seedlings we had. With some species, we started to seed from the best clones and later started breeding by crossing candidates with good characteristics which we wanted to combine. From around 40 species when we started in 1984, only 10 have shown potential to serve as crops for the future, and a few of them are already grown commercially. It was surprising to discover that it is not enough to develop knowledge to grow and handle the good clones, but that marketing R&D is also necessary to attain commercial success.

In this review, I shall describe some of the successes and failures we have had in our R&D efforts since 1984.

## 2. Wild Fruit Trees Which We Domesticated

The best success we have had so far is with pitayas, a common name for cacti, which produce fruits on long shoots. We introduced around 12 of these species, two of which excelled as follows:

Vine cacti species from the genera ***Hylocereus*** and ***Selenicereus*** known today as pitaya or "dragon fruit" were introduced from the wild (from Yucatan in Mexico and other southern American countries, up to Colombia). We also obtained genotypes from botanical gardens and cacti amateurs (see list descriptions and pictures @ URL: http://www.bgu.ac.il/life/mizrahi.html). The first clones we released

to growers were not tasty but beautiful, and the growers encouraged us that they would sell them as ornamental fruits. Organoleptic taste tests were performed, where 1 is the lowest and 5 is the best. These first clones obtained scores between 1 and 2, while our new hybrids obtain taste scores between 3.5 and 4.5. Vietnam grew these cacti with the name "Dragon fruit" and were the first to sell them in the world market as early as 1994. By 1995, we sent our first clones to Europe. It took us much R&D effort to develop the necessary knowledge to grow and ship them. We have a huge collection of genotypes, the characteristics of which we have described elsewhere (see: http://www.bgu.ac.il/life/mizrahi.html). We found that some of them were tasty, and it was possible to get hybrids from different clones, species, and even genera [7]. Today, we have tasty fruits with good horticultural characteristics grown in Israel and elsewhere. We can provide summer and winter clones of different hybrids of pitaya fruits almost year-round. When we started to work on these genotypes, there was almost nothing in the scientific literature. In a few years' time, the number of publications started to grow on an exponential scale (Figure 1).

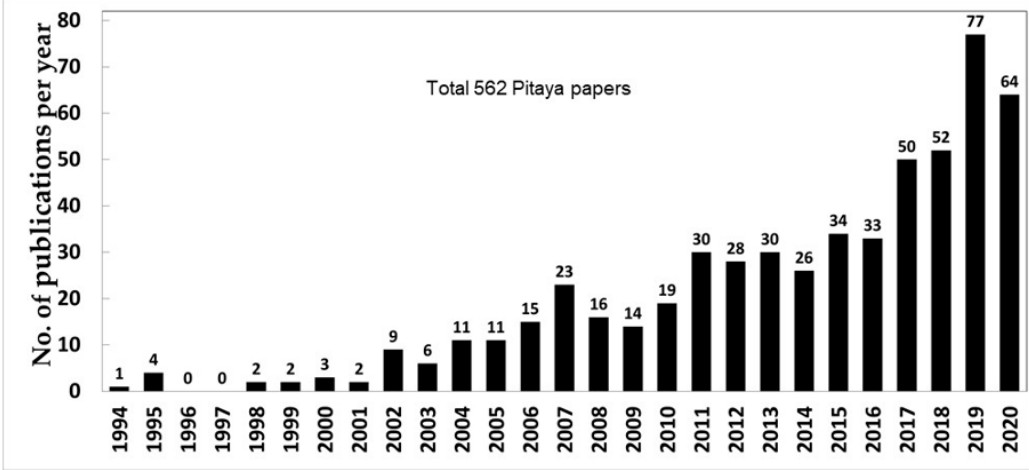

**Figure 1.** Number of scientific publications per year on pitayas. Taken from Web of Science.

Today, there are more than 30 countries growing and marketing pitayas worldwide (Haj Yihye and Yael Kahal. Special report—potential of pitaya export, Department of Marketing Ministry of Agriculture State of Israel in Hebrew).

Figure 2 shows the differences in appearance of various genotypes even among clones of the same species.

In 2014, we published a summary of our knowledge on pitaya [8]. All the genotypes of these genera need to be grown in Israel in the shade, since the Israeli full sunlight causes photoinhibition [5,9]. Today, there are around 30 growers in Israel selling pitaya fruits, and this number is growing due to their high profitability.

We also domesticated another species from the Cactaceae family, known around the world as the ornamental plant "Princess of the Night" *Cereus peruvianus*. Since its reproductive biology was not known, it was not possible to obtain its fruit. We found that it is a self-incompatible genotype and cross-pollination is required to obtain fruits [10]. The same was true for the above-mentioned pitayas known as the ornamental cactus "Queen of the Night" [11]. Since all fruits of columnar cacti are known as pitaya, we invented the name **Koubo** for the ***Cereus peruvianus*** to distinguish it from the vine cacti, which is known as pitaya [12]. We found that it is crossable with the Mandacaru *Cereus jamacaru* that is known in Brazil. In 2014, we published a summary of our results on Koubo [13]. The first grower to grow it gave up, since he was unable to sell this unknown fruit. Then, another grower realized that this new crop has excellent characteristics as follows: it uses 10% of the water required by $C_3$ and $C_4$ plants (Table 2).

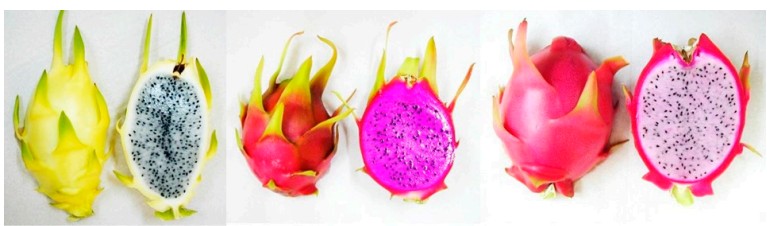

*Clones of H. undatus*

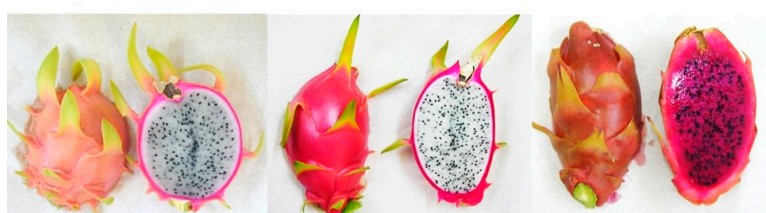

**Figure 2.** Fruits showing great differences among the clones of one pitaya species *Hylocereus undatus.*

**Table 2.** Horticultural water use efficiencies (HWUE) of common fruit crops and fruit species with the CAM photosynthesis pathway. Data collected from growers in the Negev Desert of Israel.

| Fruit Crop | Yield Tons/Hectare | Irrigation $m^3$ Water Hectare$^{-1}$Year$^{-1}$ | HWUE, Tons of Water/Ton Fruit |
|---|---|---|---|
| **C3 Crops** | | | |
| Pear—*Pyrus communis* | 15 | 6870 | 458 |
| Peach—*Prunus persica* | 12 | 6280 | 523 |
| Avocado—*Persea americana* | 12–20 | 9440 | 786–472 |
| Various Citrus—*Citrus* sp. | 35–80 | 10,000–12,000 | 150–285 |
| **CAM Crops** | | | |
| Koubo—*Cereus peruvianus* | 25 | 1200–1600 | 48–64 |
| Opuntia—*Opuntia ficus-indica* | 30 | 2500 | 83 |
| Vine cacti—*Hylocereus* and *Selenicereus* spp. | 35 | 1200–1600 | 34–46 |

The fruit is red, with a shiny smooth red peel, white pulp, and excellent aromatic taste. The yield is around 20 tons/hectare/year. The fruit shelf life is over a month, and in Israel so far no pests have appeared to infest this crop (Figure 3).

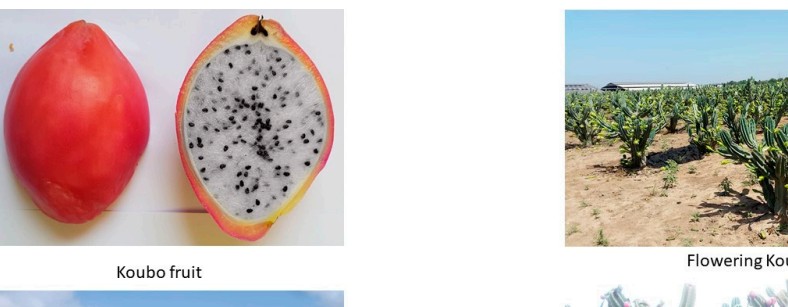

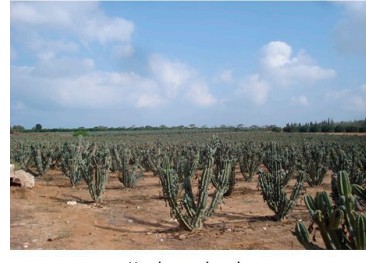
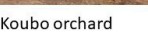

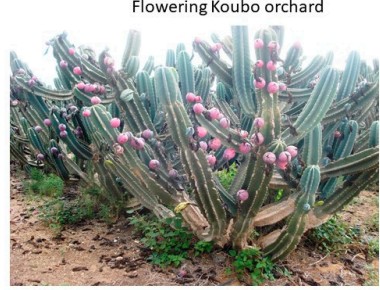

**Figure 3.** View of Koubo orchard, 4 years old flowering orchard, ripe fruits ready for picking, and open Koubo fruit.

The grower took the fruits to a big supermarket in Israel and gave them out to consumers to taste. Those who liked them purchased them, and he built the market for this new fruit. Today, he and other growers who have joined him are selling around 140 tons/year in the domestic market. Due to its profitability and the market demand, more growers have started to grow it. Several consumers in Europe have already showed an interest in the Koubo. This was the first time we realized the importance of marketing new crops. The **Cactaceae** family is ideal to work with in this project, since they have the best Water Use Efficiency (WUE) known within regular crops (Table 2) [14].

From the Anacardiaceae, we domesticated the Marula, *Sclerocarya birrea* **sbsp.** *caffra* [15]. The germplasm came from Botswana as seedlings and a few bud woods of clones from the late Professor Holtzhauzen L.C. of Pretoria University, South Africa. From the very beginning, we realized that this plant could grow very rapidly (2 m a year) and produce several hundreds of Kg per tree per year, even under the harsh conditions of the Negev Desert (Figure 4).

The plant even managed to grow at Neot Hakikar on the shores of the Dead Sea, where the summer temperatures reach over 40 °C and the water is saline (E.C. = 4 dS/m), as well as under similar conditions in Qetura, in the Arava valley [4,16]. On top of the stress tolerance advantage of this species is the fact that there is no need to pick it from the tree. When mature, the fruits abscise as green, after around a week turn ripe and yellow in color, and are then ready for consumption as an aromatic delicious fruit. We exposed this fruit to several organoleptic taste tests and over 85% people graded it as having an excellent taste. For many years, no one wanted to invest in this multipurpose fruit. However, the indigenous people of Southern Africa were consuming it fresh, using its nuts, extracting excellent oil out of it, and producing various products from it. Recently, a company located in the Israeli Negev desert started to collect fruits (this year around 30 tons) from our seedlings and grafted orchards and started to produce various products. This is the first real attempt to use our knowledge for the commercial production of Marula fruit and its products.

Another African wild fruit tree we domesticated is the Argan *Argania spinosa* from the *Sapotaceae* family [17]. It is an endemic species to Morocco, where the local people extract oil from its seeds. It has a reputation for its excellent oil, and we obtained support for this in our research [18,19]. This project started by collecting the seeds from 130-year-old trees grown in the Mikveh-Israel Botanical garden near Tel-Aviv. We found a huge variability among the seedlings. Some died due to *Fusarium oxysporum* infection, others produced very little oil, and a few others gave us yields over 1 Kg oil per tree per year extracted from its seeds. One major obstacle to its domestication was the difficulty of propagating

the high-oil-yielding trees by vegetative methods. In the last two years, two nurseries and the lab at the Agriculture Research Organization (ARO, Volcani Center) have started to propagate this tree by cuttings, and the first orchards were planted recently with several selected clones. We expect that in 4 years' time we shall be able to analyze this species' profitability as a new oil crop. The advantage of this species is the fact that it can be grown under a range of temperatures, from minus 6 to 45 °C. It can tolerate the Ca, Mg, and $SO_4$ salinity (E.C. = 4 dS/m) which exists in Qetura, but not the E.C. = 4 dS/m of Na and Cl salinity which exists in Neot Hakikar and Ramat Negev. Fruit harvesting can be mechanized, since the fruits drop when ripe. Rootstocks should be developed to tolerate *Fusarium oxysporum*, which killed around 10% of the seedlings. This pathogen is found in most Israeli soils, and it is not clear whether this fungus is the same genotype that attacks olives, almonds, and avocados. In addition, we found that cross-pollination is an advantage for obtaining high yields—namely, mixing clones in the orchard [20].

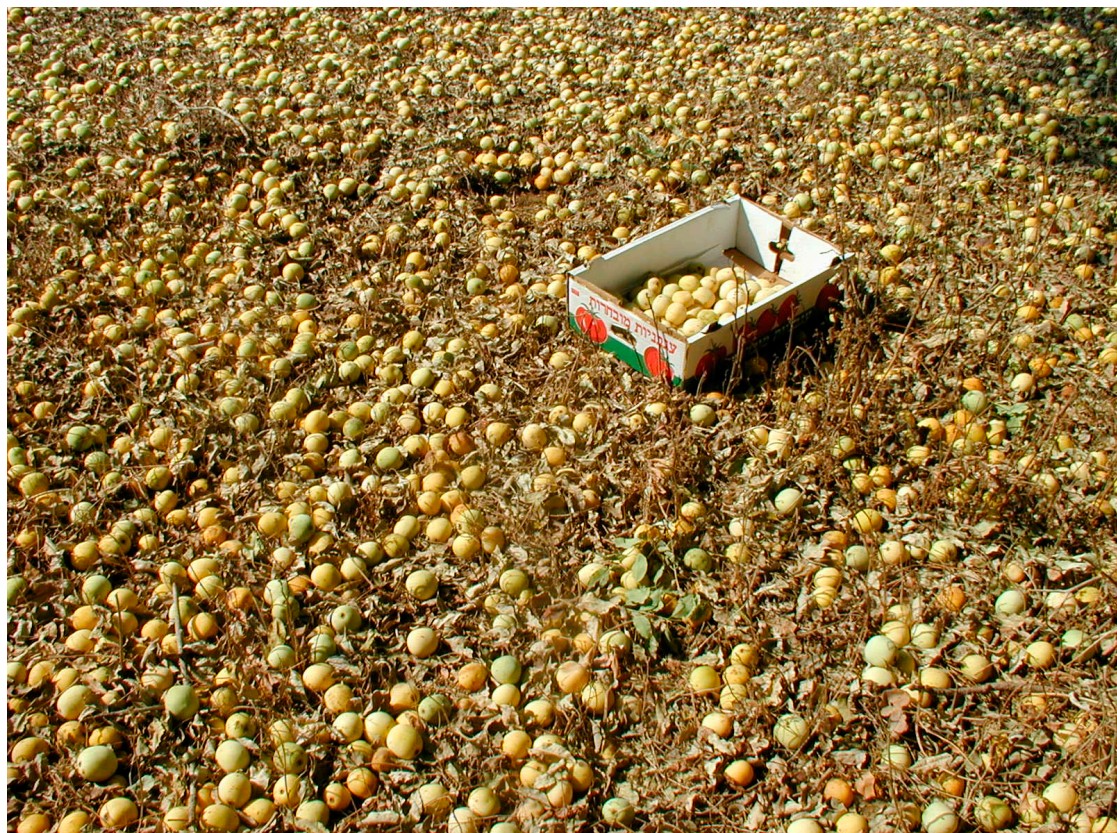

**Figure 4.** Abscised fruits from the Marula tree. Note green and yellow fruits. Yields are high, at a few hundred Kgs/tree/year.

## 3. Exotic Fruit Trees We Developed

We experienced several failures along the way; among them is the White sapote—***Casimiroa edulis***—from the *Rutaceae* family. This is a fruit tree grown locally in Mexico and Guatemala. Small fruit quantities can be found in specialty shops in the USA and Australia, but this fruit is largely unknown around the world. We obtained seeds from amateur growers of exotic fruit trees both from Israel and from the USA. We obtained the bud wood of few clones from a collection in the Agricultural Research Organization (ARO) the Volcani Center. They grew very well even under low salinity [21] and started to produce in about five years in seedlings and two years in grafted trees. The fruits look like green apples, have a smooth sweet melting pulp, and were very popular with tasters. However, around 50% of the tasters claimed that the fruits had a bitter aftertaste. For the further development of this crop, we selected only clones which had no bitterness (selected clone shown in Figure 5).

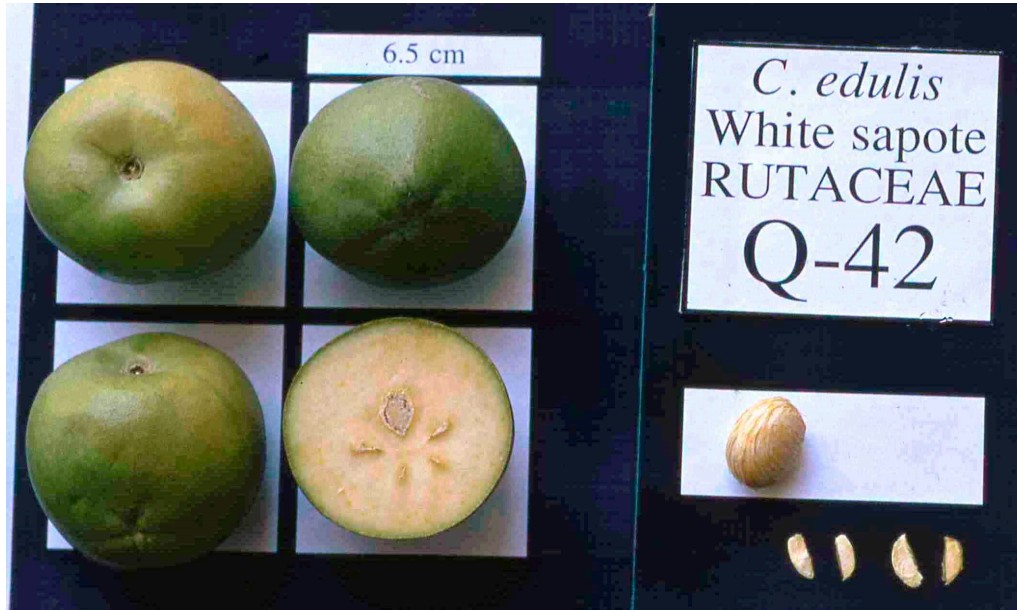

**Figure 5.** Appearance of the selected white sapote clone with a small number of developed seeds.

Among the seedlings, we found four groups of clones with different flower types as follows: hermaphrodite flowers, flowers with a normal gynoecium but small androecium, staminate flowers (female sterile with no stigma), and pistillate flowers (male sterile with no pollen sacs). The flower type is important when growing seedlings in order to select a superior clone. This clone must be a hermaphrodite or otherwise pollinizers need to be planted among the single clone orchard. The white sapote has three waves of flowering; in autumn, in winter, and in the spring. In Israel, the wave that produces most of the fruits is the winter flowering. The flowers are visited by a huge number of honeybees and are recommended as a honeybee crop. One grower was very happy with this species and wanted to plant a commercial orchard. We suggested that he test first the acceptability of this new fruit in the local market. He obtained from us 5 tons of good tasty clones with instructions on how to bring them in prime conditions to the market. Unfortunately, he was unable to sell even one single fruit. Even the merchants who specialize in exotic fruit refused to give it a chance. This was the second time that we realized that the bottleneck for such a project is marketing. Those who would like to work on such projects need marketing R&D, such as Israel carried out in the 1950s when the avocado was introduced to Europe.

We also had a marketing failure with ***Ziziphus mauritiana*** from the *Rhamnaceae* family (also known as Indian jujube). We found it to be an excellent crop for harsh desert conditions. It tolerated temperatures between minus 6 °C and 45 °C and 4 dS/m salinity, both Na, Cl and Ca, Mg and $SO_4$. We found it to be equivalent in salt tolerance to date palms. The fruit are used and sold fresh and/or dry, and are similar in appearance to dry dates. On other hand, my colleague the late Professor Dov Pasternak successfully introduced this species as a new fruit tree crop to the Sahel in Africa and named it Pomme Du Sahle [22]. Today, it is used as a new fruit for the indigenous people of this desert region.

## 4. Conclusions

Thousands of plant species exist around the world with the potential of being future crops. In Australia, they have active discussions on how to potentially cultivate a long list of species candidates [3] (also see The Australian New Crops Newsletter—electronic resource).

Unfortunately, very little research has been performed on potential new crops in academic institutions around the world. This may stem from the fact that agricultural granting agencies require any research proposal to answer the questions: "How much of this new crop will be sold? For what

price?" Since no one can give these figures, they usually refuse to support such proposals. This was our experience.

If we want to be prepared to face the problem of global warming, we must give priority to supporting new crop research proposals. It is also very important to provide funds for marketing R&D activity, in parallel to all R&D efforts invested in agriculture and the biological aspects of developing future crops. From our R&D activity, we succeeded with the vine cacti pitaya, Koubo, Marula, and Argan. We do have gene banks of various clones we selected, waiting for investors to try. These fruit trees species are white sapote—*Casimiroa edulis*; black sapote—*Diospyrus digyna*; Indian jujube—*Ziziphus mauritiana*; monkey orange—*Strychnos spinosa*; and sapoldilla—*Manilkara sapote*.

**Conflicts of Interest:** The authors declare no conflict of interest.

# Appendix A

**Table A1.** Full list of our candidate species.

| Family<br>Botanical Name | Common Name | Distribution |
|---|---|---|
| **Apocynaceae** | | |
| *Carissa grandiflora* A. DC. | Carrisa | Southern America |
| **Anacardiaceae** | | |
| *Sclerocarya birrea* subsp. *caffra* Sounder | Marula (Morula) | Southern Africa |
| *Spondias cytherea* (*Spondias dulcis*) Forst | Ambarella | Polynesia |
| **Bombacaeae** | | |
| *Bombax glabra* | Malabar nut | Central America |
| **Cactaceae** | | |
| *Acanthocereus tetragonus* (L.) Humlk. | Acanthocereus | Mexico |
| *Cereus peruvianus* (L.) Miller | Apple cactus (Pitaya) | North South America |
| *Escontria chiotilla* (Weber) Britt & Rose | Pitaya (Jiotilla) | Mexico |
| *Hylocereus costaricensis* (Weber) Br. & R. | Pitahaya | Central America |
| *Hylocereus paolyrhi* (Weber) Br. & R. | Pitahaya | Central America |
| *Hylocereus polyrhizus* (Weber) Br. & R. | Pitahaya | Central America |
| *Hylocereus purpusii* (Weber) Br. & R. | Pitahaya | Central America |
| *Hylocereus undatus* (Weber) Br. & R. | Pitahaya | Central America |
| *Myrtilloactus geometrizans* (Mart.) Cons. | Pitaya | Mexico |
| *Nopalea cochenillifera* (L.) Salm-Dyck | Nopalito, Nopalea | Mexico |
| *Opuntia ficus-indica* (L.) Miller | Prickly pear | Tropical America |
| *Opuntia streptacantha* Lem. | Prickly pear | Tropical America |
| *Pachycereus pringlei* (Berger) Britt & Rose | Cardon pelon | Sonoran Desert |
| *Selenicereus megalanthus* (Schum.) Br. & R. | Pitaya | Columbia |
| *Stenocereus griseus* (Haw.) Buxb. | Pitaya | Oaxaca Mexico |
| *Stenocereus gummosus* (Engelm.) Gilbs. | Pitaya agria | Sonoran Desert |
| *Stenocereus stellatus* (Pfeiff.) Riccob. | Pitaya | Mexico |
| *Stenocereus thurberi* (Engelm.) Buxb. | Pitaya dulce | Sonoran Desert |
| *Stenocereus thurberi* var. *litoralis* (E.) B. | Pitaya dulce | Sonoran Desert |
| **Caesalpiniaceae** | | |
| *Cordeauxia edulis* Hemsl. | Yehib | Northeast Africa |
| **Ebenaceae** | | |
| *Diospyros digyna* Jacq. | Black sapote | South America |
| *Diospyros discolor* Willd. | Mabolo (Velvet apple) | Philippine Islands |

**Table A1.** *Cont.*

| Family<br>Botanical Name | Common Name | Distribution |
|---|---|---|
| *Diospyros mespiliformis* Hocht. | Mmilo namibia | South Africa |
| **Euphorbiaceae** | | |
| *Ricinodendron rautanenii* Schinz | Mongongo | Southern Africa |
| **Guttiferae** | | |
| *Rheedia madruno* Triana & Planch. | Madrono | Central America |
| **Flacourtiaceae** | | |
| *Dovyalis caffra* Warb. | Kei apple | Southern Africa |
| **Leguminosae** | | |
| *Tamarindus indica* L. | Tamarind | Tropical Africa |
| **Loganiaceae** | | |
| *Strychnos cocculoides* Backer | Monkey orange | Southern Africa |
| *Strychnos spinosa* Lam. | Monkey orange | Southern Africa |
| *Strychnos pungens* Solereder | Monkey orange | Southern Africa |
| **Mimosaceae** | | |
| *Inga* spp. | Ice cream bean | South America |
| **Moraceae** | | |
| *Artocarpus heterophyllus* Lam. | Jackfruit | Asia |
| **Rhamnaceae** | | |
| *Ziziphus mauritiana* Lank. | Ber | Old World Tropics |
| **Rosaceae** | | |
| *Prunus salicifolia* H BK. | Capulin cherry | Mexico |
| **Rubiaceae** | | |
| *Vangueria infausta* Burch. | Mmilo | Southern Africa |
| **Rutaceae** | | |
| *Casimiroa edulis* Llave & Lex. | White sapote | Mexico, Central America |
| **Santalaceae** | | |
| *Santalum acuminatum* (R. Br.) A. DC. | Quandong | Australia |
| **Sapotaceae** | | |
| *Argania spinosa* L. | Argan | Morocco |
| *Chrysophyllum cainito* L. | Star apple | Central America |
| *Manilkara zapota* van Royen | Sapodilla | India, Africa, Central America |
| *Mimusops angel* Engler | Angel | Somalia |
| *Mimusops zeyheri* Sond. | Mmupudu | Southern Africa |
| *Pouteria sapota* (Jacq.) Merr. | Mammey sapote | Central America |

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
