# Peer review of "Do We Need New Crops for Arid Regions? A Review of Fruit Species Domestication in Israel"

_agronomy, doi:10.3390/agronomy10121995_

Round 1
Reviewer 1 Report
The article brings important contributions on fruit production in arid region.
It is necessary to make some corrections to the article before publication.
Requests for changes to the article are attached to the document.

Reviewer 2 Report
This review collates the efforts over the last 30+ years to domesticate novel fruiting species for arid Israeli regions. Some species were successfully domesticated whilst other failed.
Some minor changes needed:
title: maybe specify it mostly refers to Israel experience with domesticating new crops
It might be the style of the author, but the review has been written with a lot of ‘I’ and ‘We’. I leave this to the editor to decide if suits the journal style.
References need to be numbered for Agronomy. Check style as journal title should be abbreviated.
Line 26: reword as same as first sentence of abstract
Lines 31 and 32: change world into global. Change large production quantities into large production volumes
Line 39: delete full stop after future
Line 47: delete or move 'possibly' as it is between 'to' and the verb
Line 55 (Table 1): needs to be formatted according Agronomy standards. Too many vertical and horizontal lines. Also add Gilgal to the Locations.
Line 57 and other parts of the manuscript where relevant: either spell ions out or abbreviate, do no have a mix
Line 59: add reference for all of the locations in Table 1.
Line 77: edit 'either too low or high' into 'extreme'
Line 141: change farmer into grower. Also present in other parts of the review. Edit accordingly
Line 111: edit 'in a logarithmic scale' to 'exponentially' as Figure 1 shows an exponential growth
Line 119: not sure if 'clones' here is the right word. Are the different images shown in Figure 2 different genotypes? If not, are they clones of the same genotypes with mutations (similar to ‘Gala’ and ‘Royal Gala’ in apple)? To me a clone is an individual propagated by vegetative means and identical to the mother plant.
Line 145: table 2: edit CAM pathway into CAM photosynthesis. Spell out HWUE in Table legend. Is table format appropriate for Agronomy?
Line 173: incomplete sentence. Add ‘very quickly’ or ‘very rapidly’ after ‘could grow’
Line 241: change ‘different structure of flowers as follows…’ into ‘different flower types as follows: hermaphrodite flowers, flowers with normal gynoecium but small androecium, staminate flowers (female sterile with no stigma), pistillate flowers (male sterile with no pollen sacs)’
Line 245: edit to ‘the flower type is important when growing seedling to select a superior clone’
Line 249: change bisexual into hermaphrodite.
Line 283: edit ‘If we want to be prepared to face the global warmi
